# Antiproliferative and Apoptotic Effects of Cardamonin against Hepatocellular Carcinoma HepG2 Cells

**DOI:** 10.3390/nu12061757

**Published:** 2020-06-12

**Authors:** Nassrin A. Badroon, Nazia Abdul Majid, Mohammed A. Alshawsh

**Affiliations:** 1Institute of Biological Sciences, Faculty of Science, University of Malaya, Kuala Lumpur 50603, Malaysia; nabtqm@hotmail.com; 2Department of Pharmacology, Faculty of Medicine, University of Malaya, Kuala Lumpur 50603, Malaysia

**Keywords:** Hepatocellular carcinoma, cardamonin, apoptosis, cell cycle arrest, NF-κB pathway, ROS production

## Abstract

Liver cancer is the sixth most common cancer in terms of incidence and the fourth in terms of mortality. Hepatocellular carcinoma (HCC) represents almost 90% of primary liver cancer and has become a major health problem globally. Cardamonin (CADMN) is a natural bioactive chalcone found in several edible plants such as cardamom and *Alpinia species.* Previous studies have shown that CADMN possesses anticancer activities against breast, lung, prostate and colorectal cancer. In the present study, the mechanisms underlying the anti-hepatocellular carcinoma effects of CADMN were investigated against HepG2 cells. The results demonstrated that CADMN has anti-proliferative effects and apoptotic action on HepG2 cells. CADMN showed potent cytotoxicity against HepG2 cells with an IC_50_ of 17.1 ± 0.592 μM at 72 h. Flow cytometry analysis demonstrated that CADMN arrests HepG2 cells in G1 phase and induces a significant increase in early and late apoptosis in a time-dependent manner. The mechanism by which CADMN induces apoptotic action was via activation of both extrinsic and intrinsic pathways. Moreover, the findings of this study showed the involvement of reactive oxygen species (ROS), which inhibit the NF-κB pathway and further enhance the apoptotic process. Together, our findings further support the potential anticancer activity of CADMN as an alternative therapeutic agent against HCC.

## 1. Introduction

Hepatocellular carcinoma (HCC) represents almost 90% of primary liver cancer and is considered the sixth most common cancer and fourth most frequent reason for cancer death worldwide [1]. Chronic infection with hepatitis B virus (HBV) or hepatitis C virus (HCV) are the top risk factors for liver cancer worldwide. People with cirrhosis have a higher risk of liver cancer [2]. There are several noticeable problems involved in the current clinical therapy of HCC, such as high recurrence rate associated with drug resistance [3]. In addition, chemotherapeutic agents exhibit many adverse effects such as cytotoxicity, fever, abdominal pain, infection, nausea, weakness and tiredness. Therefore, intensive pharmacological studies have been carried out in order to find an alternative anticancer agent which is safe and more effective to treat HCC [4,5,6]. Cardamonin (CADMN) is a chalconoid that has been extracted from cardamom spice and found in many plants including *Alpinia gagnepainii*, *Boesenbergia rotunda*, *Piper hispidum* and other edible plants [7]. CADMN showed cytotoxic activities against an array of cancer cell lines including A549 (lung), DU145 (prostate), MDA-MB-231 (breast), MCF-7 (breast), U266 (myeloma), CCRF-CEM (leukemia), and SGC7901 (gastric) [8]. Furthermore, CADMN has been shown to reduce tumor growth in mice [8], however there are limited studies on the effect of this compound on HCC. Previous studies have revealed that CADMN exerts its anticancer activity through alteration of various pathways such as mTOR, STAT3, Wnt/β-catenin and NF-κB signaling pathways [8]. The aim of this study is to investigate the antiproliferative and apoptotic action of CADMN against HepG2 hepatocellular carcinoma (HCC) cells and in addition, to elucidate the underlying molecular mechanisms at the protein level.

## 2. Materials and Methods

### 2.1. Compounds

Cardamonin (CADMN) was obtained from Sigma Aldrich, USA with molecular weight 270.28 g/mol and purity > 98% and dissolved in DMSO (0.02%) for in vitro work. 5-Fluorouracil (5-FU) was obtained from MP Biomedical, lllkirch, France and dissolved in DMSO (0.02%). All other chemicals were purchased from Sigma and Fisher with analytical grade.

### 2.2. Cell Lines

Two cell lines were used in this study, namely HepG2 human HCC cells which were derived from the liver tissue of a 15-year-old American adolescent boy of European ancestry with a well-differentiated hepatocellular carcinoma and Hs27 human fibroblast cell line. Both cell lines were obtained from American Type Culture Collection (ATCC, Manassas, VA, USA). HepG2 cell line was cultured in EMEM media and Hs27 cells were cultured in DMEM media, both media containing 1% penicillin/streptomycin, 10% fetal bovine serum and maintained at 37 °C incubator with 5% CO_2_.

### 2.3. Cell Proliferation MTT Assay

The in vitro cytotoxic effect of CADMN was determined by using the MTT colorimetric assay which is a microculture tetrazolium salt (MTT, Sigma, St. Louis, MO, USA) as described by Mosmann [9]. In brief, cells (5 × 10^4^ cells/well) were treated with different concentrations of CADMN or 5-FU and incubated for 24 h, 48 h and 72 h. Then, 20 μL of MTT solution (5 mg/mL) was added to each well and the plate was re-incubated for 4 h. Then, 100 μL of DMSO was used to dissolve the formazan crystals. The absorbance was measured with a microplate reader (Tecan, Infinite M1000) at 570 nm. 5-FU was used as a positive control and drug of reference in this experiment. The inhibition effect of compounds was performed in triplicates and expressed as IC_50_ value. The cell inhibition percentage was estimated as follows:$$\%\;\mathrm{Cell}\;\mathrm{Death}=\frac{\mathrm{OD}\;\mathrm{sample}\;-\;\mathrm{OD}\;\mathrm{control}}{\mathrm{OD}\;\mathrm{control}}\times 100$$

### 2.4. Cell Cycle Analysis

HepG2 cells were cultured in a 25 cm^2^ culture flask at 10^6^ cells density in a complete medium. Cells were treated with the IC_50_ of CADMN for 24 h, 48 h and 72 h. Experiment was performed in triplicates. DMSO (0.02%) was added to the untreated cells. Then cells were harvested, washed with cold phosphate buffered saline (PBS), and fixed at 4 °C in 70% ethanol overnight. After that, cells were centrifuged for 5 min at 2000 rpm, resuspended in PBS, and incubated in 100 µg/mL propidium iodide and 50 µg/mL RNase A (Becton Dickinson, San Jose, CA, USA) at room temperature in the dark for 30 min. Cells were analyzed to determine the distribution of the cell cycle phase using a BD FACS Canto II flow cytometer (Becton–Dickinson, San Jose, CA, USA) [10].

### 2.5. AO/PI Assay (Acridine Orange/Propidium Iodide Double Staining Assay)

Morphological alterations induced by CADMN in HepG2 cells were determined using AO/PI acridine orange/propidium iodide double staining assay [11]. In brief, HepG2 cells were cultured in a 25 cm^2^ culture flask at 10^6^ cells density in a complete medium and treated with CADMN for 24 h, 48 h and 72 h. After that, cells were harvested and washed with PBS and cell suspensions were mixed with staining solution 10 µL (1:1) containing 10 μg/mL propidium iodide and 10 μg/mL acridine orange (dissolved in PBS). Green shrinking cells condensed of fragmented nucleus (apoptotic), green intact cells (the viable) and red cells (necrotic) are the morphological changes in HepG2 cells that were observed using fluorescence-inverted microscope (Nikon, Japan) within 30 min [11]. The experiment was repeated three times.

### 2.6. Annexin V/PI Assay

Analysis of cellular apoptosis of untreated and treated HepG2 cells was investigated using Annexin V-FITC assay (BD Biosciences, San Jose, CA, USA) according to the manufacturer’s instructions. This procedure was performed in 6 mL polystyrene round-bottom FACS tubes. The optimum concentration was 2 to 4 × 10^6^ cells per 200 μL volume for flow cytometry analysis. Cells were treated with the IC_50_ of CADMN at different time points (24, 48 and 72 h). The experiment was repeated three times for each time point. The cells were washed two times with cold PBS, then Annexin binding buffer (1×) was added at the concentration of 1 × 10^6^ cells/mL. Then 5 µL Alexa Fluor Annexin V conjugate and PI (5 µg/mL) was added to (100 µL) cell suspension and incubated in the dark for 30 min. After that, binding buffer (400 µL) was added to each tube and analyzed by FACSCalibur flow cytometer system (Becton Dickinson, San Jose, CA, USA) [12].

### 2.7. Caspase 3/7, 8 and 9 Measurements

Caspase 3, 7, 8 and 9 are members of the proteases family and they are the core mediators of the apoptosis leading to cell death [13]. Caspase-Glo 3/7, 8, and 9 assays are the most sensitive caspase assay kits available in the market (Promega, Madison, WI, USA). Briefly, HepG2 Cells were seeded in white plate (SPL, Korea) at 1 × 10^4^ density of cells per well and incubated at 37 °C overnight. Then cells were treated with the IC_50_ of CADMN for 24 h, 48 h and 72 h. Then cells were incubated for 1 h in dark at 25 °C with 100 µL of caspase Glo reagent contains substrate, buffer and MG-123 inhibitor. All samples were mixed using plate shaker for 30 s at 300–500 rpm, and the measurement of the caspase activities was done by a luminescence microplate reader (Tecan Infinite, Männedorf, Switzerland). This assay was performed three times for each time point.

### 2.8. Measurement of Multi-Parameter Cytotoxicity Using Confocal Microscope

Multi-parameter cytotoxicity assay was performed with Cellomics multiparameter cytotoxicity 3 kit (Thermo Scientific™, Pittsburgh, PA, USA) according to the instructions of the manufacturer. This assay was performed to measure cell health-independent parameters, which include cell loss nuclear size, morphological changes, changes in cell membrane permeability, mitochondrial outer membrane potential (MOMP) and cytochrome C release [14]. Cells were treated with the IC_50_ of CADMN and after that cells were stained with suitable dye or antibody, then analyzed by confocal microscope. The chemotherapeutic drug 5-FU was used as a positive control in this assay. Briefly, HepG2 cells were seeded in 12 well-plate covered by a cover slide for 24 h at 37 °C. Then cells were treated with CADMN for 72 h and stained with YoYo dye (Life Technologies, Camarillo, CA, USA) for cell permeability and MitoTracker dye (Life Technologies, Camarillo, CA, USA) for MOMP detection. After that, cells were blocked with 1X blocking buffer after being fixed, and permeabilized. Then, cytochrome C primary antibody probes were added to the cells for 1 h. After that, the secondary DyLightTM 649 conjugated goat anti-mouse IgG was added for another 1 h. Nucleus of CADMN-treated cells were stained with Hoechst 33342. Cover slides were removed from the plat wells and placed on the slide. After that, stained cells were rinsed gentility with PBS and introduced to a confocal Leica TCS SP5 II microscope (Leica Microsystems, Mannheim, Germany). Intensity measurements were detected using Image J software (National Institutes of Health (NIH), Bethesda, MD, USA) [14].

### 2.9. Protein Expression Analysis (Proteomics)

Proteome Profiler Apoptosis Array Kit (R&D Systems, Minneapolis, MN, USA) was used to determine the expression levels of 32 different apoptotic proteins according to the instructions of the manufacturer. Briefly, HepG2 cells were treated with the IC_50_ of CADMN for 72 h and lysed using a lysis buffer. Protein concentrations were quantified via the Bradford method [15]. Blocking solution reagent and cell lysates protein were incubated overnight with the membranes at 4 °C. After that, a reconstituted detection antibody cocktail was added to the membranes and incubated for 1 h after the membrane was washed. Then, the membranes were washed and incubated for 30 min with streptavidin horseradish peroxidase-conjugated. Finally, after Chemiluminescent reagents were added, the membranes were exposed to X-ray film to estimate the protein expression levels of CADMN-treated and untreated cells. The intensity of the arrays was analyzed by Image J software (NIH, Bethesda, MD, USA). The repeated spots intensity was averaged and then standardized using the reference spots.

### 2.10. Detection of NF-κB Activity

NF-κB is a transcription factor which regulates many physiological processes including cell death, inflammation, cell proliferation and immune response [16]. NF-κB protects the cells from undergoing apoptosis as a response to DNA damage or cytokines. Regulation of the signaling pathways of NF-κB activity plays an essential role in cancer development and progression [16]. NF-κB kit (Thermo Fisher Scientific, Pittsburgh, PA, USA) was used for this assay. Briefly, HepG2 cells were seeded overnight at a density of 4 × 10^4^ in 12 well-plates covered by a cover slide. Then cells were treated with the IC_50_ of CADMN for 24 h, 48 h and 72 h and then induced by 100 ng/mL TNF-α (Santa Cruz, CA, USA) for 24 h. After that, cells were fixed and stained according to the protocol provided by the manufacturer and analyzed using confocal Leica TCS SP5 II microscope (Leica Microsystems, Mannheim, Germany). The Intensity of cytoplasmic and nuclear NF-κB was measured using Image J software (NIH, Bethesda, MD, USA). The average intensity of 35 objects/sample was quantified. The ratios were then compared among TNF-α-stimulated, CADMN-treated, and untreated cells [14]. 5-FU was used as a positive control.

### 2.11. Measurement of Reactive Oxygen Species (ROS) Generation (DNA Damaging)

The fluorescent probe, DCFH-DA (2,7-dichlorofluorescin diacetate), (ab113851, DCFDA Cellular ROS Detection Assay Kit, Abcam) was used to detect the production of intracellular ROS [17]. Briefly, 2.5 × 10^4^ HepG2 cells were seeded in 96-well black plates. The following day, cells were incubated for 45 min with 25 μM DCFH-DA which diluted in serum-free medium at 37 °C in darkness. Then, after removing the cell medium and washing cells in FBS-free media, 100 µL/well of different concentrations of CADMN (IC_50_ (17.1 μM), IC_70_ (54.4 μM) and IC_90_ (157.9 μM) at 72 h) and IC_50_ of 5-FU were added and incubated for 120 min. Then, fluorescence was measured at 485 nm excitation and 535 nm emission using a fluorescence microplate reader (Tecan Infinite M 200 PRO, Männedorf, Switzerland).

### 2.12. Statistical Analysis

Data were articulated as a mean ± standard deviation. Significance was analyzed by one-way analysis of variance (ANOVA), followed by post hoc for comparing the tested groups to control group. *p* < 0.05 was considered as statistically significant. Data were analyzed with graph pad prism, version 5 for windows and SPSS Statistic 20 (SPSS Inc., Chicago, IL, USA).

## 3. Results

### 3.1. Cardamonin Inhibits Cell Proliferation of HepG2 Cells

The cytotoxic effect of CADMN against human HCC cell line HepG2 and normal fibroblast cells Hs27 was examined by MTT colorimetric assay. CADMN and 5-FU significantly inhibited the growth of HepG2 cells in a dose- and time-dependent manner (Figure 1a,b). As shown in Figure 1c, the half-maximal inhibition concentration (IC_50_) of CADMN-treated HepG2 cells at 24, 48 or 72 h was 307.6 ± 131.7 μM, 217.1 ± 35.7 μM and 17.1 ± 0.592 μM, respectively. 5-Fluorouracil was used as a positive control and exhibited a more pronounced cytotoxic effect against HepG2 cells with an IC_50_ of 256.7 ± 76.87 μM, 85.4 ± 4.50 μM and 14.6 ± 4.61 μM after 24, 48 and 72 h of treatment, respectively (Figure 1d). The IC_50_ of CADMN-treated HepG2 after 72 h was significantly lower than the IC_50_ of CADMN after 24 h and 48 h (Figure 1c). Therefore, the IC_50_ of 72 h was picked to be investigated in the downstream assays including the molecular mechanism. On the other hand, CADMN showed high selectivity and less cytotoxic effect on Hs27 normal fibroblast cells with an IC_50_ 225.7 ± 15.53 μM at 72 h as compared to 5-FU (IC_50_ 11.53 ± 3.075) (Table 1).

### 3.2. Cardamonin Induces HepG2 Cells Arrest at G1 Phase

Cell proliferative inhibition can occur through two major pathways: cell cycle arrest or apoptosis induction. To determine whether CADMN influences cell cycle progression, flow cytometry was used to demonstrate the distribution of CADMN-treated HepG2 cells at different cell cycle stages. As shown in Figure 2, CADMN treatment extraordinarily caused HepG2 cells to be accumulated in G1 phase in a time-dependent manner. The cell cycle analysis of CADMN-treated HepG2 cells at different time points (24, 48 and 72 h) revealed a higher cells number in G1 phase as compared to untreated cells. Thus, the result indicated that CADMN treatment at 72 h effectively mediated G1 cell cycle arrest in HepG2 cells.

### 3.3. Cardamonin Alters HepG2 Cell Morphology

AO/PI staining of CADMN-treated HepG2 cells clearly shows morphological features of apoptosis which increased in a time-dependent manner (Figure 3). The AO/PI stained images presented some changes in CADMN-treated HepG2 cells in terms of morphology and color, which indicate an apoptosis induction. The untreated control cells show intact cells stained with a green color representing the viable cells. A few greenish-orange cells with cell membrane blebbing were observed in CADMN-treated HepG2 cells after 24 h, which indicate an early stage of apoptosis. CADMN-treated HepG2 cells for 48 h revealed late stage of apoptosis which was demonstrated by more blebbing and the color turned to reddish-orange. Upon 72 h of treatment, the orange-stained cells were changed to red and apoptotic bodies were clearly indicating cell death. The orange-stained HepG2 cells increased time dependently, as well as more morphological changes appeared after 48 and 72 h of CADMN treatment, suggesting an alteration from viable cells to cell death via apoptosis in a time-dependent manner.

### 3.4. Cardamonin Induces Apoptosis in HepG2 Cells

In order to determine whether the cell growth inhibition by CADMN was accompanying the induction of apoptosis in HepG2 cells and to confirm the quantitative efficiency of apoptosis induction, CADMN-treated HepG2 cells and untreated cells were stained with Annexin V-FITC and PI. Flow cytometry analysis was used to distinguish the cells according to the cell staining and their different phases (viable: PI and Annexin-V negative, early apoptosis: PI negative and Annexin-V positive, late apoptosis: PI and Annexin-V positive, and necrotic: PI positive and Annexin-V negative). Figure 4 shows that 78.7% of cells were viable in untreated cells, while viable cells reduced in a time-dependent manner to 65.3%, 59.9% and 37.8% of viable cells at 24 h, 48 h and 72 h, respectively. Moreover, CADMN-treated HepG2 cells at different time points (24, 48 and 72 h) showed an increment of early and late apoptosis in a time-dependent manner which peaked at 31.9% and 35.6%, respectively, after 72 h treatment.

### 3.5. Cardamonin Induces Caspase-3/7, -8 and -9 Activities

Apoptosis is categorized into caspase-independent or caspase-dependent mechanisms. To investigate the exact mechanism during the apoptosis process in CADMN-treated HepG2 cells, the bioluminescent intensities of caspase-3/7, -8, -9 activities at 24, 48 and 72 h were measured using a caspase cascade kit. As shown in Figure 5, there was a significant increase in caspase-3/7, -8 and -9 activities in CADMN-treated HepG2 cells in a time-dependent manner. Thus, these data indicated that apoptosis is triggered through both intrinsic and extrinsic pathways in CADMN-treated HepG2 cells.

### 3.6. Multiparameter Cytotoxicity Analysis

The multi-parameter cytotoxicity assay kit was used for this assay. HepG2 cells were exposed to CADMN to investigate five different measurements including nuclear intensity, cell count viability, cell membrane permeability, cytochrome C release and mitochondrial outer membrane permeabilization (MOMP). Images of untreated HepG2 cells and treated cells with CADMN or 5-FU were also captured by confocal microscope (Figure 6a).

#### 3.6.1. Cardamonin Induces Alterations in MOMP and Cell Membrane Permeability

Reduction of Mitotracker dye (red) in CADMN-treated HepG2 and 5-FU treated cells indicates mitochondrial damage and increase in mitochondrial outer membrane permeabilization (MOMP) compared to the control untreated cells (Figure 6b). HepG2 cells treated with CADMN and 5-FU for 72 h showed significantly less fluorescence intensity which reflected the mitochondrial membrane damage. In addition, a significant increase in cell membrane permeability was shown by increment in the intensity of YoYo dye (green) in 5-FU and CADMN-treated cells indicating an alteration in cell membrane permeability (Figure 6c).

#### 3.6.2. Cardamonin Induces Release of Cytochrome C to Cytosol and Nuclear Condensation

Cytochrome C stained weakly in untreated cells. In contrast, 5-FU and CADMN treated-HepG2 cells were stained strongly (Figure 6d). This suggests that CADMN and 5-FU induced the leakage of cytochrome C into cytosol from the mitochondria in HepG2 after 72 h treatment. In addition, markedly increased nuclear condensation (Hoechst 33342, blue) was detected in 5-FU and CADMN-treated HepG2 cells as compared to untreated cells (Figure 6a).

### 3.7. Protein Expression Analysis (Proteomics)

Determination of the expression levels of apoptotic and anti-apoptotic proteins was investigated on CADMN-treated HepG2 cells at 72 h. The results showed significant upregulation of pro-apoptotic proteins such as FADD, FAS, TRIAL, HIF-1 and significant increase in phosphorylation of p53, which indicate the activation of intrinsic and extrinsic apoptotic pathways. Moreover, the results showed significant increase of caspase 3 cleavage/pro-caspase 3 ratio in CADMN-treated cells as compared to the untreated HepG2 cells, which is common in caspase in both apoptotic pathways. On the other hand, there were a significant downregulation of anti-apoptotic proteins such as heat shock proteins (HSP60, HSP27 and HSP70), XIAP, catalase, clusterin, and survivin (Figure 7).

### 3.8. NF-κB Translocation Suppressed by Cardamonin

Nuclear factor kappa B (NF-κB) is a transcription factor that regulates many important cellular processes such as inflammatory responses, cellular growth and apoptosis. Translocation of NF-κB from cytoplasm to nucleus leads to apoptosis inhibition. In untreated HepG2 cells, TNF-α stimulation resulted in an increase in NF-κB translocation from the cytoplasm to the nucleus (Figure 8). 5-FU and CADMN significantly reduced the TNF-α-stimulated NF-κB translocation and suppressed the NF-κB translocation into the nucleus in a time-dependent manner (24, 48 and 72 h). Thus, results suggest the involvement of a NF-κB inhibitory mechanism in the apoptosis process.

### 3.9. Cardamonin Induces High Level of ROS

Chemical or environmental stress on cells could lead to generate reactive oxygen species (ROS) as a result of oxygen metabolism, which is followed by cytoskeletal structure modification and cell apoptosis. The ROS accumulation levels in 5-FU and CADMN-treated HepG2 cells was assessed by staining cells with DCFH-DA dye and the fluorescence intensities were measured with a fluorescence microplate reader. The levels of DCF fluorescence in CADMN-treated HepG2 cells were significantly increased in a dose-dependent manner (Figure 9). CADMN and 5- FU after 2 h of treatment triggered ROS production in HepG2 cells significantly as compared to untreated cells.

## 4. Discussion

Chemotherapy is an important strategy for treatment of HCC, however high cytotoxicity of chemotherapeutic agents and low selectivity may cause severe damage to the normal cells [18,19]. Moreover, chemotherapy is accompanied by negative side effects such as neurotoxicity, cardiotoxicity, hepatotoxicity, nephrotoxicity, myelosuppression and gastrointestinal reaction. Therefore, intensive efforts should be directed to identify potential anticancer agents with a better safety profile and wider therapeutic window by exploring phytoconstituents and bioactive compounds isolated from natural products [20,21]. Chalcones either of a natural or synthetic source have been reported as a potential anticancer agents against several types of cancer [22].

CADMN is a chalcone that has gained a great interest by researchers because of its potential anticancer and antioxidant activities [23]. Previous studies showed that CADMN exhibits anticancer activities against various kinds of cancer including lung, prostate, colorectal carcinoma, breast, nasopharynx, multiple myeloma, ovarian glioblastoma and colorectal carcinoma [24,25,26,27]. Moreover, CADMN showed cytotoxic activities and antiproliferative effects against an array of cancer cell lines including A549 (lung), U266 (myeloma), DU145 (prostate), CCRF-CEM (leukemia), MDA-MB-231 (breast), MCF-7 (breast), and SGC7901 (gastric) cells. In addition, it has been reported that CADMN reduces tumor growth in mice [8].

The mechanisms underlying the anti-cancer action of CADMN against HCC have not been studied in detail. Therefore, in this study we investigated the anticancer activity of CADMN in HepG2 human hepatocellular carcinoma cells and identified the possible mechanistic pathways of the anticancer activity of CADMN.

The current study showed that CADMN exhibited both time- and dose-dependent anti-proliferative effects against HepG2 hepatocellular carcinoma cells, whereas having less cytotoxicity against normal fibroblast Hs27 cell lines as compared to 5-FU. The inhibitory effect of CADMN on HepG2 cell proliferation after 24 h and 48 h was obvious but was more pronounced and significant after 72 h of the treatment. CADMN after 72 h of treatment exhibits a significant cytotoxic effect against HepG2 hepatocellular carcinoma cells with IC_50_ 17.1 ± 0.592 μM, which is comparable to the positive control 5-FU (IC_50_ 14.6 ± μM). Previous studies have reported the antiproliferative effects of other chemotherapeutic agents such as sorafenib, a targeted therapy for HCC, and revealed that sorafenib inhibits HepG2 cells with an IC_50_ between 7.4 and 2 μM [28,29], which is much lower than that of CADMN and 5-FU.

On the other hand, the IC_50_ of CADMN against HepG2 after 72 h was 17.1 ± 0.592 μM which is significantly lower than the CADMN cytotoxicity effect on Hs27 normal fibroblast cells (IC_50_ 225.7 ± 15.53 μM) at 72 h, which could indicate the high selectivity of CADMN. The selectivity of CADMN in HepG2 cells also indicates that CADMN is safer with low cytotoxicity on normal cells as compared to chemotherapeutic drug 5-FU. The results of cell viability suggested that CADMN has a selective inhibitory effect against HepG2 cancer cells. Similar to these results, previous studies demonstrated that CADMN has a cytotoxicity effect on breast cancer cell lines (MDAMB-231) with an IC_50_ 12.32 ± 2.11 µg/mL and 3.97 ± 1.69 µg/mL at 24 and 48 h, respectively [26], and on the human prostate cancer cell (PC-3) with IC_50_ 11.35 μg/mL at 48 h [30].

Cancer is predominantly considered as a cell cycle deregulation disease. Cell cycle progression has multiple check points which have a function to regulate the size of cell, signals of extracellular growth and DNA integrity [11,31]. Cancer can be caused by abnormal expression of positive regulators or negative regulators of the cell cycle process, which may result in abnormalities of cell proliferation. Accordingly, trigging of cell cycle arrest in cancer cells is one of the essential cancer treatment strategies. Thus, natural bioactive compounds with the ability to arrest cancer cell cycle are gaining extensive attention and are considered as an important cancer treatment strategy [11]. The accumulation of CADMN-treated HepG2 cells in G1 phase was clearly observed after 72 h, which is in agreement with the results of the MTT assay. It is suggested that growth inhibition of CADMN-treated HepG2 cells may resulted from the prevention of cell division. Contrary to these findings, previous studies showed that CADMN arrests cell cycle of SW-480 (human colon cancer cells), DLD-1 (human colorectal adenocarcinoma cells), HCT116 (human colon cancer cells) and LS174T (human colon adenocarcinoma cells) at G2/M phase [32].

According to previous studies, p21 binds to cyclin A/CDK2, E/CDK2, D1/CDK4 and D2/CDK4 complex. Induction of p21 by p53 upon DNA damage inhibits cyclin E/CDK2 and thereby inhibits G1/S transition [33]. The effect of CADMN on expression of p21 and p53 was evaluated by Proteome Profiler Apoptosis Array Kit. The results showed upregulation in the expression of both p21 and p53 proteins. Thus, suggesting that CADMN induced G1 arrest through regulation of p21, p53 and Cyclin D1 [33]. Another study demonstrated that CADMN exerted a dose-dependent cytotoxic effect on human nasopharynx carcinoma cell line (NPC) after 24 h of incubation and arrested cell cycle at G2/M [25].

The important feature of apoptosis is cell shrinking, causing condensation of organelles and cytoplasmic density, which appears during the early stage of apoptosis. The most important feature of early apoptosis is chromatin condensation while budding, which involves wide blebbing of the plasma membrane associated with very tight packed organelles, is the feature of late apoptosis. Our data showed that there were characteristic apoptotic morphological alterations in the HepG2 cells after 24 h of CADMN treatment, but after 72 h of treatment cell necrosis was more obvious as compared to 24 h. Moghadamtousi et al. reported that treatment with CADMN for a longer time led to increased occurrence of necrotic cells [11].

Moreover, to confirm these results quantitatively, Annexin-V/PI double stain assay was performed using flow cytometry analysis. Many biochemical modifications, involving breakdown of DNA, protein cross-linking and cleavage of the proteins cause the morphological changes in apoptotic cells. The appearance of cell surface markers such as phosphatidylserine (PS) is considered as one of those biochemical modifications. To investigate the biochemical characterization of apoptosis, an Annexin-V-FITC assay was carried out to examine the externalization of phosphatidylserine (PS), which is translocated to the outer plasma membrane in the early apoptosis. PI was used to detect the late apoptosis and necrosis, whereas Annexin-V was used for the early and late apoptosis detection [11]. CADMN treatment at various time points (24, 48 and 72 h) led to time-dependent increment of early and late apoptosis rate which peaked at 31.9% and 35.6%, respectively, after 72 h treatment.

Caspases are a family of protease enzymes with specific cysteine protease activity playing a role in controlling the degradation of cellular components in order to minimize its effect on surrounding tissues during the cell death process. Caspases enzymes are divided into two major groups, executioner (caspase-3, -6, -7) and initiator (caspase-2, -8, -9, -10) caspases.

Extrinsic and intrinsic apoptosis pathways are activated by caspase-8, and -9, respectively. Both initiator caspases lead to executioner caspase-3 or -7 activation, which prompt cell apoptosis. Treatment of HepG2 cells with CADMN for different time periods (24, 48 and 72 h) resulted in a time-dependent activation of caspase-3, -8, and -9. Previous studies supported these findings, which showed that the anticancer effect of CADMN was indicated through increase in caspase-3, -8 and -9 activities on the prostatic cancer (DU145) cell line [27].

In the intrinsic pathway, the mitochondrial membrane potential is loosed, which leads to release cytochrome C from mitochondria to cytosol. In cytosol, cytochrome C binds to Apaf-1 and in the presence of ATP it activates caspase-9, resulting in activation of caspase-3, -6, and -7 which plays a role in the cell death program as downstream effectors.

The extrinsic pathway involves death receptors such as Fas receptors, tumor necrosis factor (TNF) receptors, and TNF-related apoptosis-inducing ligand (TRAIL) receptors which bind with their specific ligands FasL, TNF-α and DR4/5, respectively. Activated complex of the death receptors and their specific ligands bind to the adaptor protein Fas-associated death domain (FADD) or TRADD (TNFR1-associated death domain protein) through their intracellular domains and activate caspase-8 by this death-inducing complex (DISC), then actives caspase-8 by cleaving initiate apoptosis directly lead to activating executioner caspase-3/6/7 [34]. Our data showed that protein expiration levels of Fas, TRAIL, FADD, DR4, DR5 and CD95 were significantly increased in CADMN-treated cells as compared to untreated cells, as well as the expression level of cleaved caspase-3. The obtained results demonstrated that CADMN induced apoptosis in HepG2 cells through the extrinsic pathways [34].

Heat shock proteins (HSPs) have anti-apoptotic functions which are related to alteration of the expression of essential apoptosis’ regulators, such as transcription factors of the nuclear factor κB (NF-κB) family, the tumor-suppressor protein p53 and pro-apoptotic members of the Bcl-2 family [19,35]. The findings of the proteomic array demonstrated that HSPs expression levels were decreased significantly in treated cells as compared to untreated cells. Moreover, clusterin was downregulated, which is known as anti-apoptosis protein.

Additionally, survivin, XIAP, cIAP1 and cIAP2 members of inhibitors of the apoptosis (IAPs) family have been shown to bind caspases and inhibit their activation, thereby preventing apoptosis [36]. In our study all IAPs were downregulated in CADMN-treated HepG2 cells.

HCC appears frequently as multiple nodules and HCC are highly vascularized. In these nodules, the tumor cells grow rapidly by scavenging an extensive amount of oxygen, and a hypoxic microenvironment results. Indeed, HCC is one of the most hypoxic tumors with a median oxygen level as low as 0.8%. During hypoxia, induction of tumor suppressor p53 may be accompanied by a hypoxia-inducible factor 1 subunit alpha (HIF1A)-dependent pathway to induce apoptosis [37]. HIF-1 can induce apoptosis through stabilization of wild-type p53, whereas it prevents apoptosis in the mutant p53 gene. HepG2 is the only HCC cell that has wild type p53 [38]. Moreover, HIF-1A can induce apoptosis via an increase in the stability of the product of the tumor suppressor gene p53. Environmental stress, DNA damage, or p53 triggers apoptosis by regulating apoptotic proteins such as Bax, or can lead to cell growth arrest by p21 [39].

Reactive oxygen species (ROS) are small molecules characterized generally by short lives and very high activity. ROS are a normal side product of the oxidative phosphorylation which occurs in the mitochondria to provide metabolic energy. Many exogenous factors such as UV, inflammatory, chemical agents and natural products have the ability to produce pro-oxidant compounds like ROS.

Previous studies reported that excessive ROS levels could result in triggering both extrinsic and intrinsic apoptotic pathways. In the extrinsic pathway, ROS activated transmembrane death receptors induce caspases-8 followed by directly triggering caspases-3/-6/-7 and activate apoptosis as aforementioned. On the other hand, ROS can activate intrinsic pathways, resulting in cytochrome C release from mitochondria and activation of apoptosis through the mitochondrial pathway. Moreover, the apoptotic mitochondrial pathway is induced by different cellular stresses, including mitochondrial DNA (mtDNA) damage. Indeed, ROS produced from the mitochondria as intracellular molecules targets mtDNA, which is susceptible to oxidative damage. Mitochondrial DNA damage would cause disruption of respiratory chain function leading to loss of mitochondrial membrane potential and destroying ATP synthesis [34]. This study showed an excessive production of ROS by CADMN and 5-FU in HepG2 cells which could result in plasma membrane and DNA damage and lead to higher membrane permeability staining and DNA fragmentation, as well as cytochrome C release as observed in the multi-parameter cytotoxicity assay [34]. Similar to these findings, one recent study showed that CADMN has an anti-proliferative effect against MDA-MB 231 and MCF-7 breast cancer cells through the induction of G2/M arrest and ROS accumulation [40].

NF-κB transcription factors play a role in hundreds of gene expression regulations that are involved in regulating many processes such as cell growth, differentiation, development, and apoptosis. It has established that the activation of NF-κB can lead to inhibiting the apoptosis process. The inhibition of NF-κB translocation could be useful in combination with conventional therapy to increase the effect of cancer therapy. NF-κB plays an important role in tumor development and progression. The correlation between NF-κB activation and oncogenesis, angiogenesis, distant metastasis, anti-apoptosis and chemotherapy resistance has previously been demonstrated. Suppressing NF-κB can lead tumor cells either to stop proliferating and then die, or to become more sensitive to the anti-tumor agents. Furthermore, NF-κB is a target in anti-cancer therapy for most of the pharmaceutical companies research [24,41]. In this study, the findings showed that the translocation of NF-κB was suppressed significantly in 5-FU and CADMN-treated cells, as well as meritoriously inhibit the activation of the NF-κB signaling pathway in HepG2 cells which suggests that the NF-κB inhibition mechanism might be involved in apoptosis.

Based on the literature, CADMN blocks the signaling of the NF-κB pathway, which was evidenced by inducing apoptosis [24,26,42]. In the present study, in vitro results showed significant inhibition of the NF-κB translocation in CADMN-treated HepG2 cells as compared to untreated cells. Thus, suggesting NF-κB involvement in CADMN anti-cancer activity.

The NF-κB pathway is closely related with ROS and accumulation of ROS can inhibit the activation of the NF-κB pathway through direct oxidation of NF-κB by ROS, which inhibits its DNA binding ability [43]. Thus, this suggests that CADMN leads to NF-κB inhibition of apoptosis through elevating the level of ROS in HepG2 cells [43]. It has been reported that CADMN arrested the cell cycle through upregulating the expression of Cyclin D1 and p21 induced by the accumulated ROS which was induced by inhibiting the activity of NF-κB. Then the accumulation of ROS led to cleavage of PARP, resulting in apoptosis [25]. Figure 10 summarizes the proposed antiproliferative and apoptotic action of CADMN on HepG2 hepatocellular carcinoma cells. CADMN activated both extrinsic and intrinsic apoptotic pathways and increased ROS level productions in HepG2 cells, which in turn involves inhibition of NF-κB translocation.

## 5. Conclusions

In our study, we demonstrated that CADMN has an anti-proliferative effect and apoptotic action on hepatocellular carcinoma HepG2 cells. In addition, CADMN led to a significant increase in early and late apoptotic cells and necrotic cells in time-dependent manner. The findings of this study also suggest the involvement of ROS which inhibits the NF-κB pathway and further enhances the apoptotic process. Together, our findings further support the potential anticancer activity of CADMN as an alternative therapeutic agent against HCC, which needs further in vivo investigations in animal models to confirm the findings.

## Figures and Tables

**Figure 1 nutrients-12-01757-f001:**
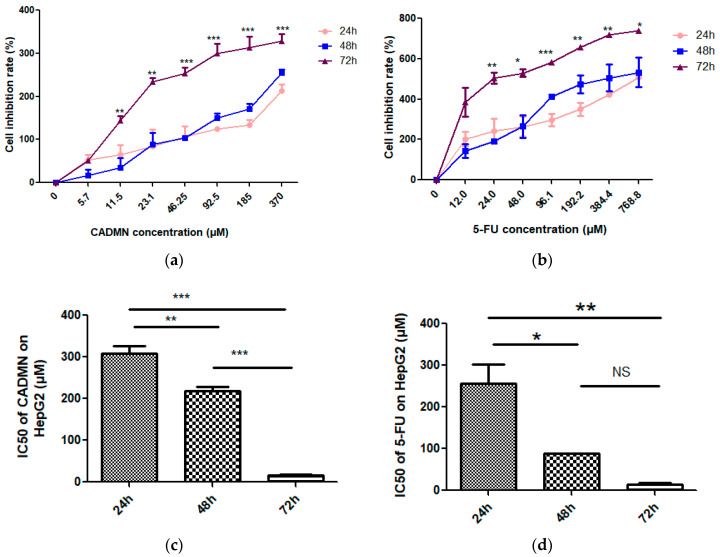
Cytotoxic effect of (**a**) Cardamonin (CADMN) and (**b**) 5-Fluorouracil (5-FU) against HepG2 cells. Different concentrations of CADMN and 5-FU were added to the HepG2 cells as a treatment for different time periods (24, 48 and 72 h). 5-FU was used as a positive control and 0.02% DMSO was used as a negative control (0). (**c**) The IC_50_ of CADMN and (**d**) The IC_50_ of 5-FU after 24, 48 and 27 h. All experiments were performed in triplicates and presented as mean ± SD. * *p* < 0.05, ** *p* < 0.01, *** *p* < 0.001 indicate significant difference.

**Figure 2 nutrients-12-01757-f002:**
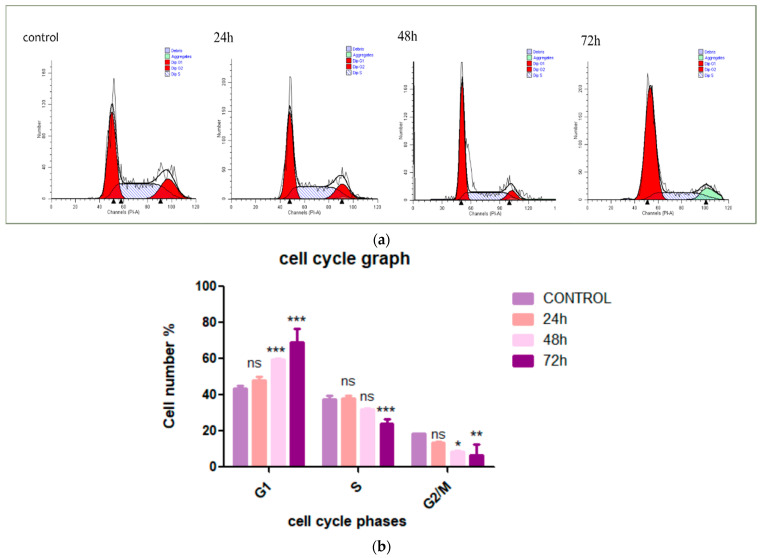
Inhibition of cell proliferation by CADMN via arresting cells at G1 phase. (**a**) Flow cytometry graphs of CADMN-treated HepG2 and untreated cells at different time points (24, 48 and 72 h). (**b**) Quantification analysis of the cell cycle assay. Data were presented as mean ± SD. * *p* < 0.05, ** *p* < 0.01, *** *p* < 0.001 indicate significant differences as compared to control.

**Figure 3 nutrients-12-01757-f003:**
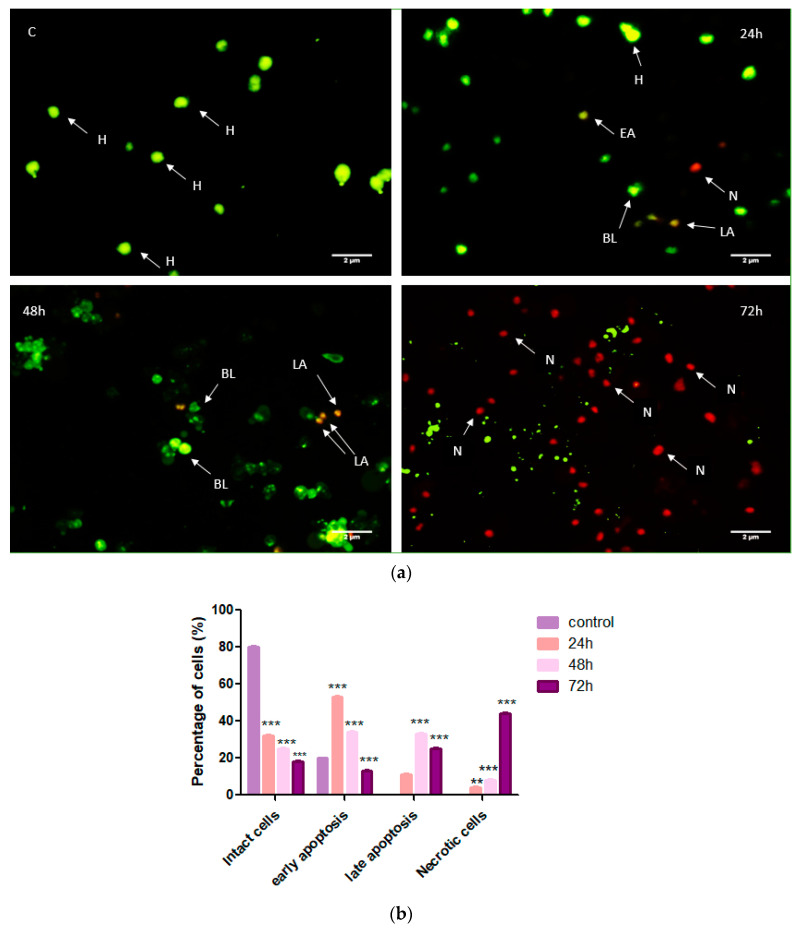
AO/PI staining of untreated HepG2 cells (control) and CADMN-treated HepG2 cells after 24, 48 and 72 h. The number of viable cells decreased in CADMN-treated HepG2 cells and cells going into the apoptosis phase steadily. (**a**) Morphological changes as observed under a fluorescence microscope (**b**) Quantitative analysis of cells percentage at different stages. Data were presented as mean ± SD. ** *p* < 0.01, *** *p* < 0.001 indicates significant differences as compared to untreated group (control). AO: Acridine orange, PI: propidium iodide, EA: early apoptotic cells, LA: late apoptotic cells, N: necrotic cells, H: intact cells, BL: blebbing.

**Figure 4 nutrients-12-01757-f004:**
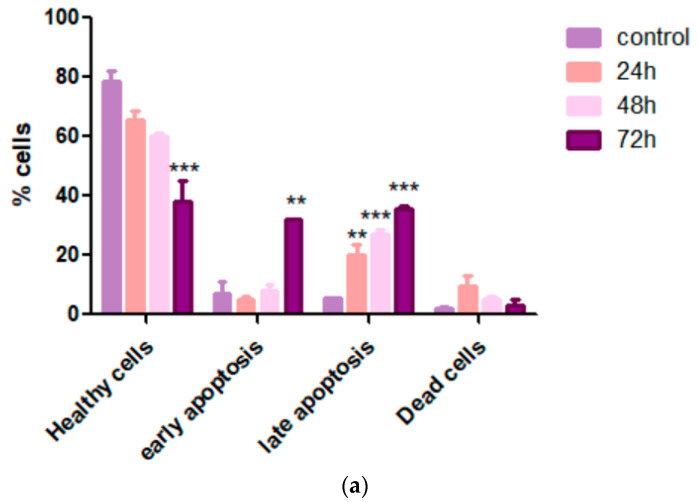
Annexin V analysis of untreated cells and CADMN-treated HepG2 cells at 24, 48, and 72 h using flow cytometry. (**a**) The calculated cells percentage at different stages. (**b**) Cells distribution at several phases. Data were presented as mean ± SD. ** *p* < 0.01, *** *p* < 0.001 indicates significant differences as compared to untreated group (control).

**Figure 5 nutrients-12-01757-f005:**
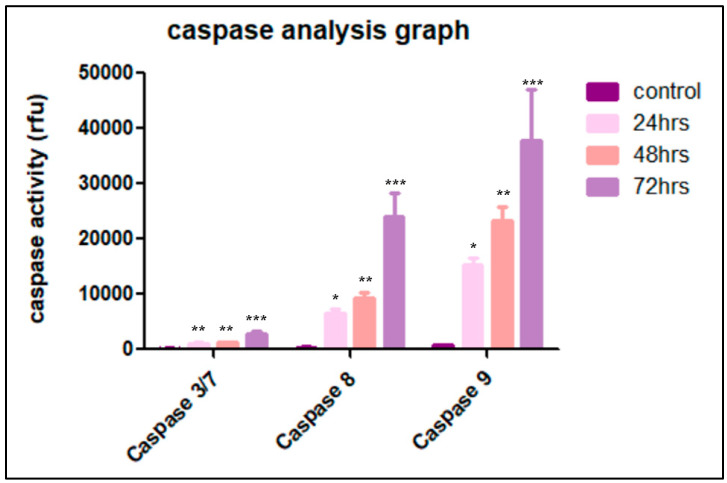
Caspases 3/7, 8 and 9 activity levels in CADMN–treated HepG2 cells estimated using Luminescence analysis at different time points of 24 h, 48 h and 72 h, as well as the control untreated cells. Data were presented as mean ± SD. * *p* < 0.05, ** *p* < 0.01, *** *p* < 0.001 indicate significant differences as compared to control.

**Figure 6 nutrients-12-01757-f006:**
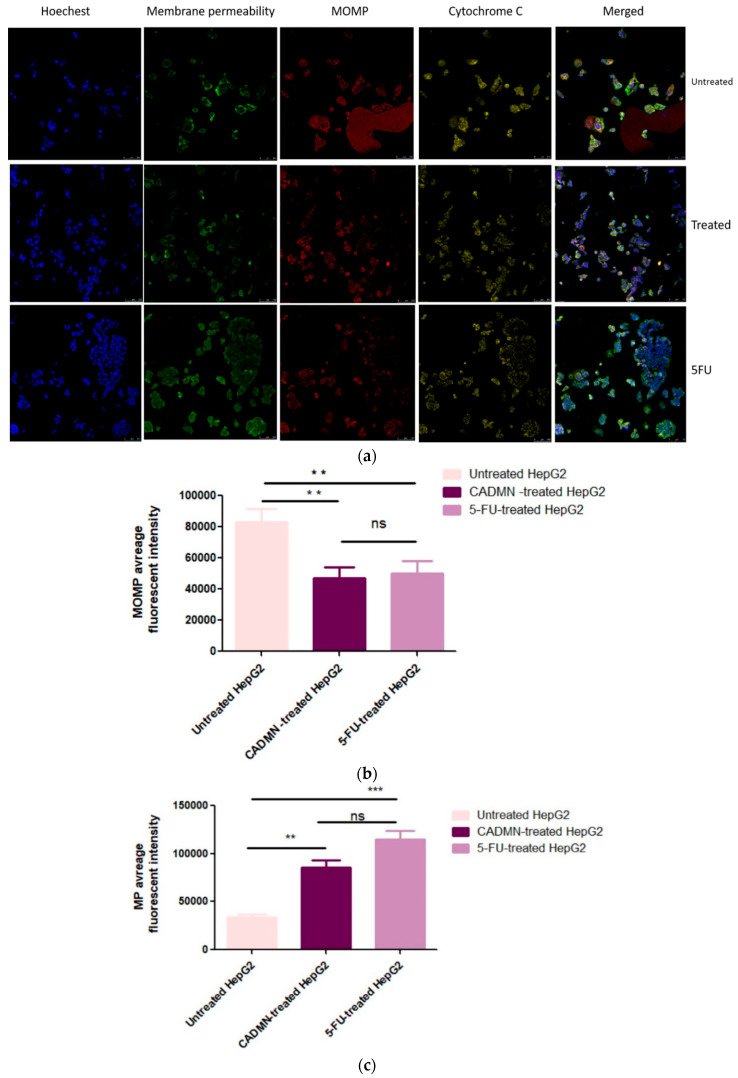
(**a**) Effect of CADMN and 5-FU on nuclear condensation (Hoechst 33342, blue) and cell membrane permeability. (YoYo dye, green), release of cytochrome C (yellow) and mitochondria outer membrane permeabilization (MOMP, MitoTracker, red) dyes. 5-FU was used as a positive control. Microphotographs were captured with a Leica Confocal microscope, magnification 20× and scale bars 100 μm. CADMN and 5-FU led to damage and increase in mitochondrial outer membrane permeabilization (MOMP) as indicated by decreases in Mitotracker dye uptake (**b**), increase in cell permeability (**c**), and leakage of cytochrome C from mitochondria to cytosol (**d**). Data were presented as mean ± SD. ** *p* < 0.01, *** *p* < 0.001 indicate significant difference as compared to control untreated cells.

**Figure 7 nutrients-12-01757-f007:**
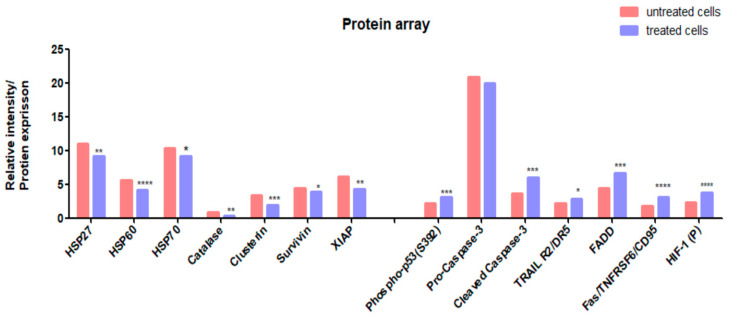
Relative expression of apoptosis-related proteins in the cell lysate of CADMN-treated HepG2 cells and untreated cells using proteome profiler array kit. Data were expressed as mean ± SD. * *p* < 0.05, ** *p* < 0.01, *** *p* < 0.001, **** *p* < 0.0001 indicate significant difference as compared to control untreated cells. HSP: heat shock proteins, cIAP-1: Cellular inhibitor of apoptosis protein 1, XIAP: X-linked inhibitor of apoptosis protein, FADD: Fas-associated protein with death domain, TRAIL R2/DR5: TNF-related apoptosis-inducing ligand receptor 2/Death receptor 5, Fas/TNFRSF6: tumor necrosis factor receptor superfamily member 6, CD95: cluster of differentiation 95, HIF-1 (P): Hypoxia-inducible factor 1-alpha.

**Figure 8 nutrients-12-01757-f008:**
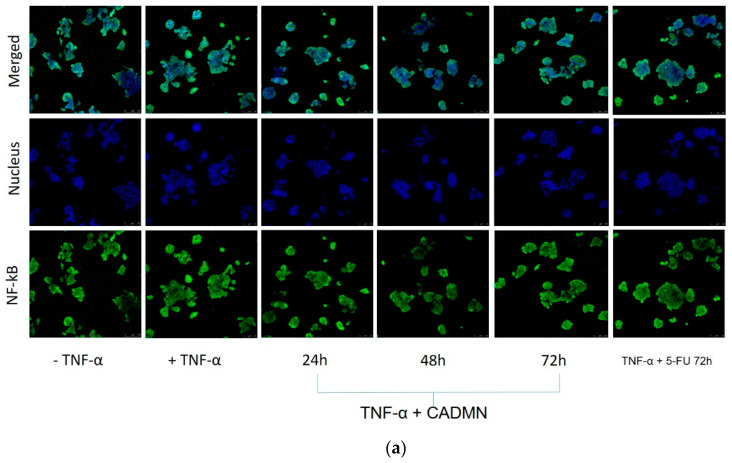
(**a**) Translocation of NF-κB was estimated in untreated cells (-TNF-α), cells treated with TNF-α (+TNF-α control), cells treated with both TNF-α and CADMN, and cells treated with TNF-α and 5-FU, Microphotographs were captured with a Leica Confocal microscope, magnification 20× and scale bars 100 μm. (**b**) Quantification analysis of nuclear NF-κB fluorescence intensity. Data were expressed as mean ± SD. *** *p* < 0.001 indicates significant difference as compared to control (+TNF-α).

**Figure 9 nutrients-12-01757-f009:**
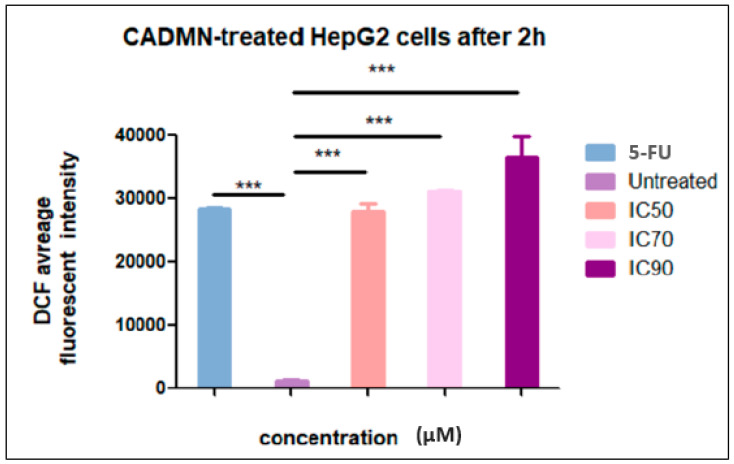
Intracellular reactive oxygen species (ROS) generation in HepG2 cells induced by different concentrations of CADMN (IC_50_, IC_70_ and IC_90_) and 5-FU (IC_50_). The fluorescence intensities were measured using a fluorescence microplate reader after 2 h of CADMN and 5-FU treatment. Results were expressed as mean ± SD. *** *p* < 0.001 indicates significantly differences as compared to control untreated cells.

**Figure 10 nutrients-12-01757-f010:**
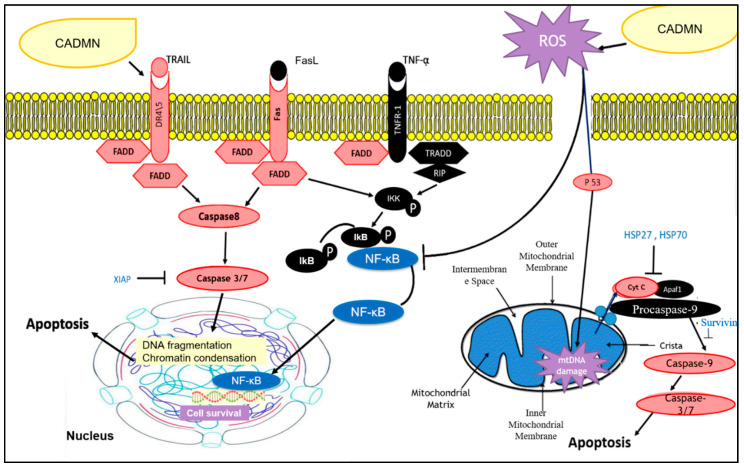
Proposed apoptotic action of cardamonin (CADMN) on HepG2 hepatocellular carcinoma cells. CADMN activated both extrinsic and intrinsic apoptotic pathways. CADMN increased ROS level production which inhibited the NF-κB pathway. Red: upregulated, Blue: downregulated, ROS: Reactive oxygen species Cyt C: Cytochrome C, NF-κB: Nuclear factor kappa B, ikk: I kappa B kinase, HSP: heat shock proteins, XIAP: X-linked inhibitor of apoptosis protein, FADD: Fas-associated protein with death domain, TNF: Tumor necrosis factor, TRAIL R2/DR5: TNF-related apoptosis-inducing ligand receptor 2/Death receptor 5, Fas/TNFRSF6: tumor necrosis factor receptor superfamily member 6, CD95: cluster of differentiation 95, HIF-1 (P): Hypoxia-inducible factor 1-alpha.

**Table 1 nutrients-12-01757-t001:** IC_50_ (µM) for cardamonin (CADMN) and 5-fluorouracil (5-FU) against Hs27 normal fibroblast cells using MTT assay at 72 h.

IC_50_ (μM)	72 h	S.I.
CADMN	225.7 ± 15.53	13.2
5-FU	11.53 ± 3.075	0.8

Data were presented as mean ± SD of three independent experiments. Selectivity Index (S.I.) gives an idea about the selectivity of compounds and the highest the value the more selectivity. S.I was calculated by dividing the IC_50_ of cardamonin-treated normal Hs27 cells by the IC_50_ of cardamonin-treated hepatocellular carcinoma HepG2 cells.

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
