# Peer review of "Antiproliferative and Apoptotic Effects of Cardamonin against Hepatocellular Carcinoma HepG2 Cells"

_nutrients, 2020, doi:10.3390/nu12061757_

Round 1

Reviewer 1 Report

The study of Badroon et al. the anticancer activity of the chalconoid cardamonin in HepG2 cells has been investigated, focusing on the possible mechanistic pathways involved.

 The following questions should be considered:

  • The used concentrations of cardamonin, as well as the values of IC50, are given in ug/ml. However, it would be advisable to make it in nano or micromolar, units that provide a better idea of the potency of the compound.
  • 5-FU is used as positive control and drug of reference. However, considering that in HCC this drug is not commonly used in the treatment of patients, it seems that this is not a good choice. So, on the one hand, the authors should explain why 5-FU has been chosen as the drug of reference and, on the other hand, they should show the effects of cardamonin in comparison to those shown by common drugs in the treatment of HCC, such as sorafenib.
  • The comparison with the effects of 5-FU is not consistent throughout the study. The authors should justify this question and show the comparative analysis with 5-FU whenever possible.
  • As a hepatic cell line of non-tumoral cells, the authors have used a human fibroblast cell line instead of healthy hepatocytes. The authors should justify this point.
  • In general, the title graphs seem unnecessary. Particularly, in figure 1 they don't keep the same style, they have text mistakes (such as mtt, instead of MTT or 5fu instead of 5-FU).
  • Figure 1 should be improved. Statistical analysis of fig 1a and 1b is lacked and the text of axes is inaccurate. Moreover, It would give more information if the data were shown by comparing in the same graph the effect of cardamonin with that of 5-FU.
  • Line 185: The sentence should be reviewed, so the authors say that cytotoxicity of cardamonin is comparable that of 5-FU. However, data show a not negligible difference in IC values (between 2.5 and 5).
  • Table 1 is redundant with Figure 1.
  • Why didn’t the authors show the effect of cardamonin in Hs27 cells on comparing with that of 5-FU? These data should be included.
  • The study of apoptotic induction through AO/IP staining, should be completed with a quantitative analysis of the percentage of cells stained of each colour.
  • Figure 5 legend should be modified. Caspases activity levels are show, not caspase levels.
  • Please review Figure 6. Use, for consistency, 5-FU treated HepG2, and include statistical analysis as comparing CADMN and 5-FU treatment in figure “c”.
  • Figure 6 legend. Please review. The effect of 5-FU is not included.
  • Regarding the results of the NF-kB translocation study, CDMN seems to reduce the TNFa-stimulated NF-kB translocation. However, the presentation of the results is not clear in that sense, and sometimes it seems not to agree with the data shown in the figure. Please, review this paragraph.
  • Figure 8 legend. Please review. The effect of 5-FU is not included.
  • English language and style changes should be improved.
  • Text editing must be extensively reviewed:
    • Please be consistent in your use of the abbreviation CADMN
    • Please review the figure legends. Several of them included sentences more appropriated in results or discussion sections.
    • Line 49: Please review this sentence. The verb seems to be missing
    • Line 185: Please correct the word Florouracil
    • Line 196: & symbol should not be used
    • Line 196: Change Liver by liver
    • Line 229: Please review this sentence
    • Line 235: Please review this sentence
    • Line 264-265: Please review this sentence
    • Line 269: Figure 5 has not the correct style
    • Line 348: Figure 5 has not the correct style
    • Line 361. Please correct the word compered
    • Line 369: Please review this sentence
    • Line 373: Please review this sentence
    • Line 376: Please review this sentence
    • Line 393: Please review this sentence. A “)” is missing

Reviewer 2 Report

This is an interesting work.

Overall comments:

  1. There are a number of syntax errors throughout the manuscript. Extensive English editing is required.

Specific comments:

Page 16 line 401-403: Cancer is predominantly considered as a cell cycle deregulation disease. Cell cycle progression has multiple checkpoints that have a function to regulate the size of the cell, signals of extracellular growth  and DNA integrity…add reference https://doi.org/10.1016/j.biotechadv.2018.11.011

Page 15 line 368-370: Chemotherapy is an important treatment procedure for patients with HCC; however, high toxicity adverse effects of anticancer agents and low specificity for tumor cells lead to consider these agents as unsafe treatment …add ref doi: 10.3390/molecules21020169

Page 4 line 178: significantly inhibits…significantly inhibited

Figure 10: The photo of the cell seems that it has been cropped from the internet. Please redraw a cell and replace it. Cardamonin and other scientific terms, for example, NF-κB and so on, are with an underline. Please remove the underlines.  Kappa B is not kB…be careful on it throughout the manuscript.

Round 2

Reviewer 1 Report

Please review the following points:

  • Figure 1: Please use the same font size in all figures
  • Figure 3: You should replace 100% with (%) in the axis title
  • Line 178: Please replace "&" by "and"
  • line 337: Please replace TNF-α-stimulation by TNF-α stimulation
